# Petrogenesis of the Nashwaak Granite, West-Central New Brunswick, Canada: Evidence from Trace Elements, O and Hf Isotopes of Zircon, and O Isotopes of Quartz

**Wei Zhang [1],\*, David R. Lentz [2]** 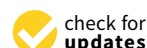 **and Kathleen G. Thorne [3]**

[1]  Collaborative Innovation Center for Exploration of Strategic Mineral Resources, School of Earth Resources, China University of Geosciences, Wuhan 430074, China

[2]  Department of Earth Sciences, University of New Brunswick, Fredericton, NB E3B 5A3, Canada; dlentz@unb.ca

[3]  Geological Surveys Branch, New Brunswick Department of Energy and Mines, Fredericton, NB E3B 5H1, Canada; kay.thorne@gnb.ca

\*  Correspondence: zhangwei-china@live.cn

**Abstract:** The petrogenesis of the Pridoli to Early Lochkovian granites in the Miramichi Highlands of New Brunswick, Canada, is controversial. This study focuses on the Pridoli Nashwaak Granite (biotite granite and two-mica granite). In situ trace elements and O and Hf isotopes in zircon, coupled with O isotopes in quartz, are used to reveal its magmatic sources and evolution processes. In the biotite granite, inherited zircon cores have broadly homogenous $\delta^{18}O_{Zrc}$ ranging from +6.7‰ to 7.4‰, whereas magmatic zircon rims have $\delta^{18}O_{Zrc}$ of +6.3‰ to 7.2‰ and $\varepsilon_{Hf(t)}$ of −0.39 to −5.10. The Hf and Yb/Gd increase with decreasing Th/U. Quartz is isotopically equilibrated with magmatic zircon rims. The biotite granite is interpreted to be solely derived by partial melting of old basement rocks of Ganderia and fractionally crystallized at the $fO_2$ of $10^{-21}$ to $10^{-10}$ bars. The two-mica granite has heterogeneous inherited zircon cores ($\delta^{18}O_{Zrc}$ of +5.2‰ to 9.9‰) and rims ($\delta^{18}O_{Zrc}$ of +6.2‰ to 8.7‰), and $\varepsilon_{Hf}(t)$ of −11.7 to −1.01. The two-mica granite was derived from the same basement, but with supracrustal contamination. This open-system process is also recorded by Yb/Gd and Th/U ratios in zircon and isotopic disequilibrium between magmatic zircon rims and quartz (+10.3 ± 0.2‰).

**Keywords:** oxygen isotopes; Hf isotopes; zircon; Ganderia; New Brunswick

## 1. Introduction

The Canadian Appalachians formed by accretion of several Gondwanan microcontinents to Laurentia as a result of Early to Late Paleozoic closure of the Iapetus and Rheic oceans [1]. Three phases of orogenesis recorded by various rock suites in New Brunswick are known as Taconic, Salinic, and Acadian orogenesis. The Taconic orogeny was manifested by accretion of three oceanic and continental terranes in the peri-Laurentia realm (500–450 Ma). Closure of the Tetagouche–Exploits backarc basin along the Bamford Brook Fault in Ganderia resulted from the Salinic orogeny (450–423 Ma). This was then followed by the Acadian orogeny due to the northwest flat-slab subduction of Avalonia beneath Ganderia along the Caledonia Fault (421–400 Ma) [2]. New Brunswick is mostly underlain by Neoproterozoic Ganderian basement (sedimentary rocks and arc volcanic rocks) that is overlain by Cambro–Ordovician, quartz-rich, passive margin sedimentary rocks. Ganderia is bordered to the north by arc volcanic rocks of the peri-Laurentian Notre Dame zone, and by the Avalonian microcontinent along the Caledonian Fault to the south [3]. Although both Ganderia and Avalonia are



peri-Gondwanan microcontinents, they have contrasting crustal compositions. Compared to Avalonia's "juvenile" Neoproterozoic crust (crustal source material with a short residence time) and ubiquitous $\delta^{18}O$-depletion of the Neoproterozoic sedimentary rocks, Ganderia has "old" Neoproterozoic basement with high-$\delta^{18}O$ sedimentary cover [4,5].

Voluminous Pridoli to Early Lochkovian granites occur in the Miramichi Highlands of Ganderia (Figure 1). Previous whole-rock Sr, Nd, and O isotope studies invoke contributions from mantle, infracrustal, and supracrustal sources in both New Brunswick and Newfoundland [3,4,6]. These granites could be associated with (a) crustal delamination after closure of the Tetagouche–Exploits backarc basin (450–424 Ma, [7]) along the Bamford Brook Fault (Salinic orogeny, [3,6]); (b) flat-slab subduction of Avalonia beneath Ganderia preceding the Acadian orogeny (423–400 Ma, [2]); or (c) other processes, such as break-off of Tetagouche–Exploits back-arc crust following the collision of Ganderia with composite Laurentia in the Late Silurian [8]. Controversy about petrogenesis of these granites needs to be clarified (e.g., [3,9,10]).

Zircon, as a refractory mineral, can potentially preserve oxygen and hafnium isotope compositions of its source as well as host magma [11–13]. Furthermore, oxygen isotopes from quartz and zircon as well as zircon trace elements are effective tools to record assimilation and fractional crystallization (AFC) processes [14–17]. Therefore, in this study we carried out in situ analysis of trace elements, Hf and oxygen isotopes of zircon, and oxygen isotopes of quartz in order to better decipher the magmatic sources of the Nashwaak Granite, as well as the physicochemical conditions during magmatic evolution, and to discuss the possible mechanism that generated the Nashwaak Granite.

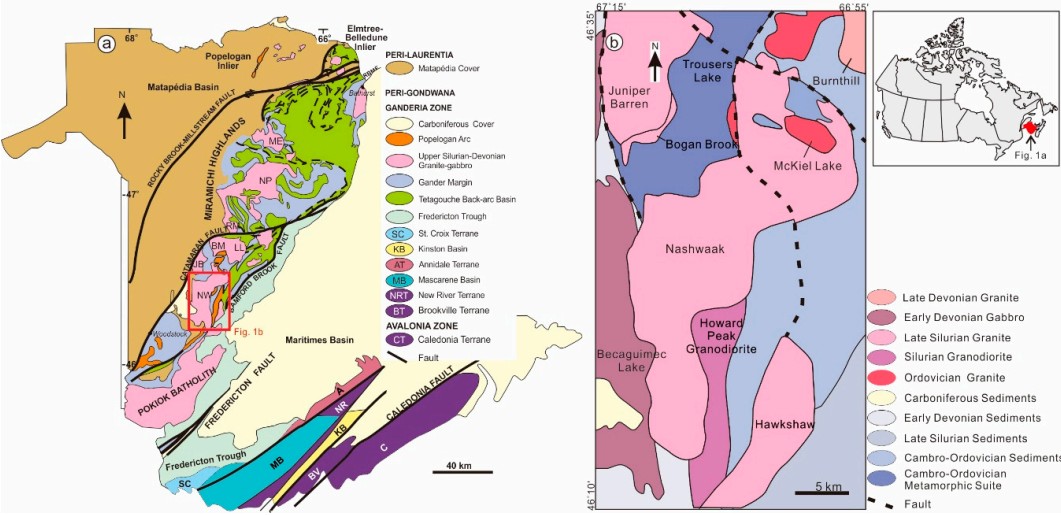

**Figure 1.** (**a**) Lithotectonic terranes of New Brunswick and (**b**) regional geological map, showing the distribution of the Nashwaak Granite (after [8]). Abbreviations of terranes: A—Annidale terrane; BV–Brookville terrane; C—Caledonia terrane; KB—Kingston basin; MB—Mascarene basin; NR—New River terrane; SC—St.Croix terrane. Abbreviations of granites: BM—Beadle Mountain Granite; JB—Juniper Barren Granite; LL—Lost Lake Granodiorite; ME—Mount Elizabeth Granite; NP—North Pole Granite; NW—Nashwaak Granitic Suite.

## 2. Geological Background

The Nashwaak Granite intruded the Cambro–Ordovician Trousers Lake Metamorphic Suite and the Ordovician McKiel Lake Granite to the north, and the Becaguimec Lake Gabbro to the west. The contact between the Nashwaak Granite and Early Devonian volcanic rocks to the south is not exposed. On its east side, the Nashwaak Granite intruded quartzose wackes, siltstones, and shales of the Cambrian–Early Ordovician Miramichi Group, and younger volcanic and sedimentary rocks of the Ordovician Tetagouche Group (Figure 1). Andalusite and cordierite are present in sedimentary rocks up to 2 km from the contact.

The Nashwaak Granite has two subfacies: (1) pink, coarse- to medium-grained, equigranular to porphyritic biotite granite with a mineral assemblage of plagioclase, orthoclase, quartz, and minor biotite, grading northward into (2) fine- to medium-grained muscovite–biotite granite containing quartz, microcline, albite, muscovite, zircon, apatite, monazite, and ilmenite. The Nashwaak Granite was formed at ca. 420 Ma and is highly siliceous (69.3–81.5 wt.%), calc-alkaline, and peraluminous, with $(La/Yb)_N$ ranging from 1.9 to 13.6, and depletion of Ba, Sr, Nb, P, and Ti. The $\varepsilon_{Nd}(t)$ is from −2.1 to −4.2 [8].

## 3. Analytical Methods

In this study, we used the same [8] zircon samples from the Nashwaak Granite as we previously dated for their U–Pb ages. Some of the zircons show inherited cores with $^{206}Pb/^{238}U$ ages ranging from 1000 to 1945 Ma and $^{207}Pb/^{206}Pb$ ages of 1040 to 2260 Ma [8]. With guidance from cathodoluminescence (CL) images, zircon grains with overgrowth magmatic rims (identified by similar $^{206}Pb/^{238}U$ age as that sample's concordia age) were selected for in situ trace element analysis at the rims. Oxygen isotope analysis was conducted on both rims and inherited cores. Only the zircon rims used for the concordia age calculation were chosen for hafnium isotope analysis. Detailed analytical methods are listed below.

### 3.1. Oxygen Isotope Analysis of Zircon and Quartz

Secondary ion mass spectrometry (SIMS) oxygen isotope analysis of zircon and quartz was carried out using a Cameca IMS 1280 ion microprobe (manufactured by CAMECA, Société par Actions Simplifiée, Gennevilliers, France) at the Canadian Centre for Isotopic Microanalysis (CCIM), University of Alberta. A $^{133}Cs^+$ primary beam was operated with an impact energy of 20 keV and a 2–4 nA beam current. The ~12 μm diameter probe was rastered slightly during acquisition to form rectangular sputtered areas of ~15 μm × 18 μm. Negative secondary ions were extracted from the sputtered area into the secondary (transfer) column by application of a 10 kV potential gradient. Transfer conditions included a 122 μm entrance slit, a 400 μm contrast aperture, and a 5 mm field aperture. The energy window utilized was 150 eV. The mass-separated oxygen isotopes were detected simultaneously in Faraday cups L'2 ($^{16}O^-$) and H'2 ($^{18}O^-$) in the multidetector array. Mass resolution (Δm/M at 10%) was typically 1950 and 2275, respectively (see [18]). For quartz, reference material S0033 was used (GeeWiz glass [19]); the measured $\delta^{18}O$ value of +12.34 ± 0.05‰ (*n* = 12) agrees well with the reported value of +12.5‰. Median uncertainties for the quartz reference materials and samples at 95% confidence (2σ) were ±0.17‰ (Table S1). For zircon, an internal reference material S0081 (UAMT1) was used; the measured $\delta^{18}O_{VSMOW}$ value of +4.81 ± 0.04‰ agrees well with the accepted value of +4.87‰ (Stern R., unpublished data). At 95% confidence, the median uncertainty for the zircon reference materials and samples is ±0.19 and ±0.18‰, respectively (Table S2).

### 3.2. Trace Element and Hf Isotopic Analysis of Zircon

The in situ zircon trace element analyses were conducted using a Resonetics M-50-LR 193 nm (manufactured by Resonetics at Kettering, OH, USA) Excimer laser ablation system coupled to an Agilent 7700× quadrupole inductively coupled plasma–mass spectrometer (ICP–MS) (manufactured by Agilent technologies at Santa Clara, CA, USA) at the University of New Brunswick. A spot size of 33 μm, a beam energy of 4 J/cm², and an 8 Hz laser repetition rate were employed. Calibration was achieved using standard procedures [20] that included use of standard reference materials (SRM) 610 glass from National Institute of Standards and Technology (NIST) for external standardization and the stoichiometric $SiO_2$ content of zircon for internal standardization (Table S3).

To evaluate whether the laser ablation spot was placed on a mineral inclusion (i.e., monazite, apatite, and titanite) that was not detectable at the scale of optical and CL imaging, the spots with extremely high content of Ca, Sr, Th, and P were carefully checked. Results with strong correlation between P and $(Sm/La)_N$, Th and $(Sm/La)_N$, or Ca/Sr and light rare earth elements (LREE), indicating the presence of monazite or apatite, were discarded (see [21]).

In situ Hf isotope analyses of zircons were conducted using a Resolution S-155 laser-ablation system with a beam size of 50 μm and a pulse frequency of 8 Hz, coupled with a Nu Plasma II multicollector ICP–MS at the State Key Laboratory of Geological Processes and Mineral Resources, China University of Geosciences (GPMR), Wuhan. The reference standard Penglai zircon with a $^{176}$Hf/$^{177}$Hf ratio of 0.282906 ± 0.000001 (2σ) was analyzed along with the samples [22]. Detailed information about the analytical method can be found in [23] (Table S4).

## 4. Results

### 4.1. Trace Elements in Zircon

Zircon rims in the two-mica granite show contents of Th and U in the range of 42 to 476 ppm and of 88 to 2260 ppm, respectively, with Th/U ratios of 0.02 to 0.91 (Figure 2a). Contents of Zr and Hf vary from 40.3 to 43.6 wt.% and 7680 to 19,080 ppm, respectively. The Eu/Eu* and (Ce/Ce*)$_D$ values calculated using the lattice-strain model of [24] are 0.08 to 1.74 and 2.65 to 25.3, respectively (Figure 2b,c; Table S3). Yb/Gd ratios in them are in the range of 9.44 to 62.0 (Figure 2d). The total REE contents range from 250 to 2487 ppm and the chondrite-normalized REE patterns show a strong enrichment of heavy rare earth elements (HREE) (Figure 3a).

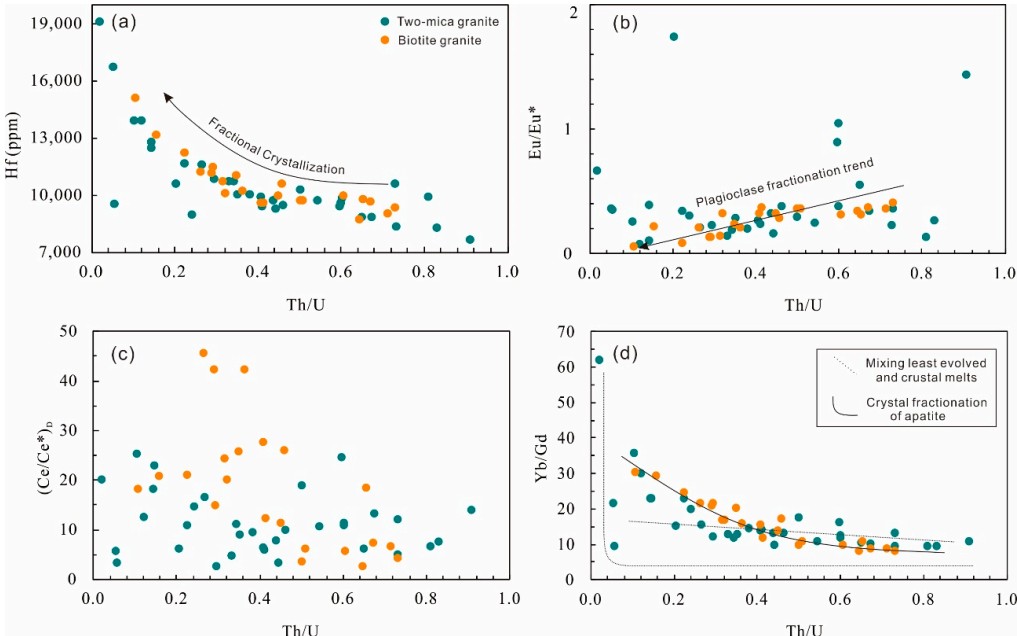

**Figure 2.** Hafnium (**a**), Eu/Eu* (**b**), (Ce/Ce*)$_D$ (**c**) and Yb/Gd (**d**) vs. Th/U of zircons from the Nashwaak Granite.

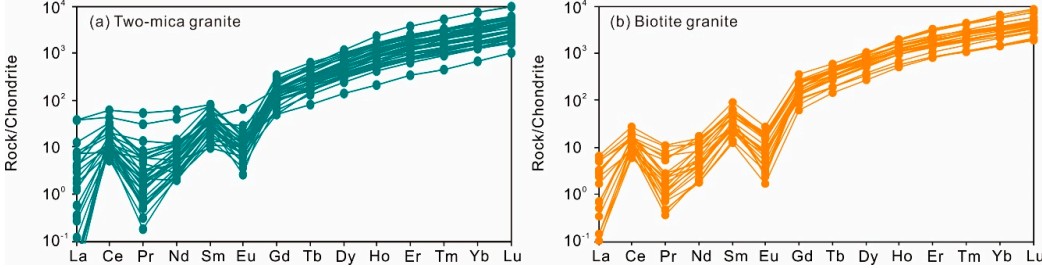

**Figure 3.** Chondrite-normalized [25] rare earth elements of zircons from the two-mica granite (**a**) and biotite granite (**b**).

The zircon rims from biotite granite show similar Th (52–363 ppm) and U (126–2480 ppm) contents and Th/U ratios (0.11–0.73) to the two-mica granite zircons (Figure 2a). Their calculated Eu/Eu* values of 0.06 to 0.41 and (Ce/Ce*)$_D$ values of 2.48 to 45.6 (Figure 2b,c; Table S3) are based on the method of [24]. They also have similar Zr (39.9–43.5 wt.%) and Hf (8750–15,090 ppm) contents. The total REE content in them ranges from 539 to 2130 ppm (Figure 3b).

### 4.2. CL Images and Oxygen Isotopes in Zircon and Quartz

The quartz grains in the biotite granite and two-mica granite are green and bluish red in ChromaSEM–CL (Figure 4). They are dominantly homogeneous and composed of only one generation of primary igneous quartz. A few quartz grains show oscillatory zoning at μm scale, revealed by small-amplitude variations in CL (Figure 4d). Secondary textures in the magmatic quartz from the Nashwaak Granite include: (1) dark CL streaks and patches associated with fractures; (2) healed fractures, which do not show up in BSE images, and are likely filled by nonluminescent SiO$_2$ (see [26]); and (3) opened fractures.

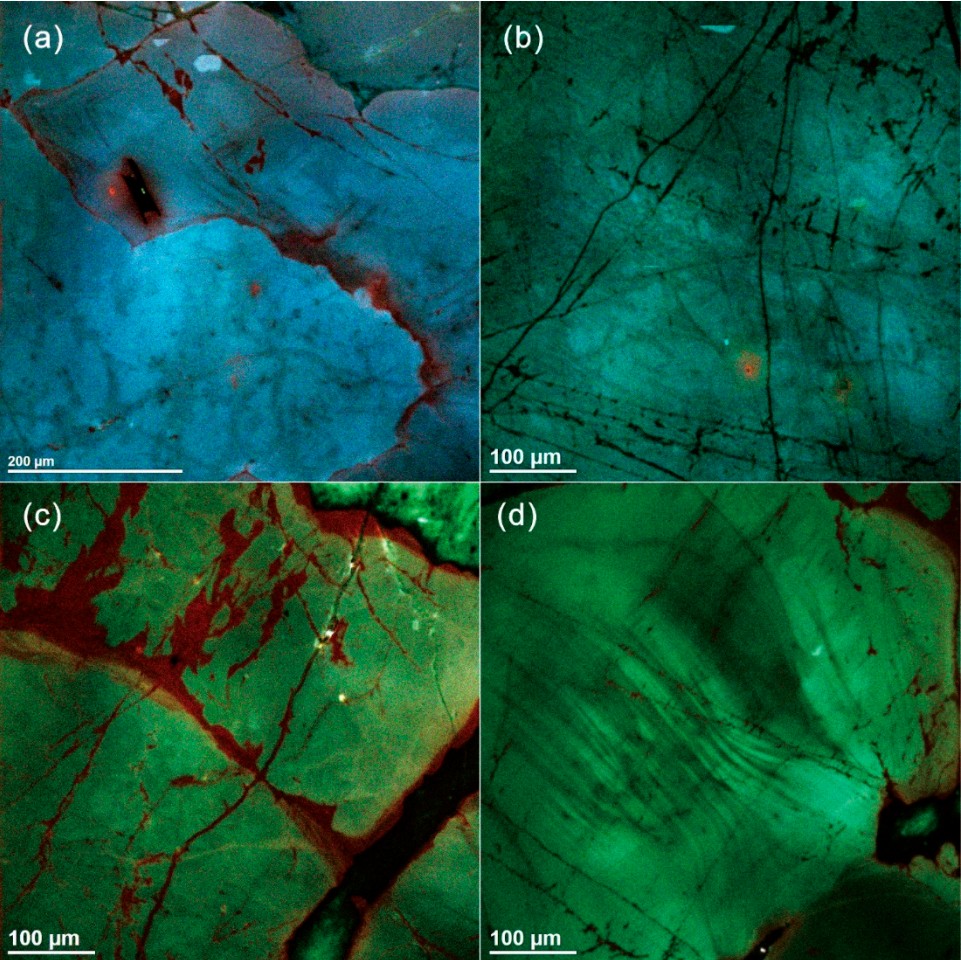

**Figure 4.** Scanning electron microscope–cathodoluminescence (SEM–CL) images of quartz from the Nashwaak Granite. In the two-mica granite, (**a**) visible oscillatory zoning parallel to grain boundary and (**b**) homogeneous quartz with fractures. In the biotite granite, (**c**) homogenous quartz with fluids infiltration along fractures and (**d**) oscillatory zoning.

Overall, $\delta^{18}$O of quartz shows limited or no intragranular variations in each granite sample. In the two-mica granite, the mean $\delta^{18}$O$_{Qz}$ is +10.3 ± 0.2‰ (2σ, *n* = 11), which is slightly higher than that of

the biotite granite with a mean of +9.7 ± 0.2‰ (2σ, *n* = 19). Detailed examination of magmatic quartz yielded consistent $\delta^{18}O_{Qz}$ values, regardless of distance from the rim of a grain (Table S1).

Two zircon populations were identified using SEM–CL: (1) prevalent high-CL response (bright) zircon domains (cores or whole crystals); and (2) thin, low-CL response rims (Figure 5). Attempts were made to analyze both of these domains; however, due to the small size of the zircon crystals and the internal fracturing observed in many grains, it was not always possible to fit multiple analyses on a single crystal. Intragrain $\delta^{18}O_{Zrc}$ values from the two-mica granite are highly variable; ranging from +5.2‰ to 9.9‰ (mean +7.4‰) for the bright cores, and +6.2‰ to 8.7‰ (mean +7.1‰) for the overgrowth oscillatory rims (Figure 6a,b). The $\delta^{18}O_{Zrc}$ values either increase or decrease from core to rim, with a range of 0.3‰ to 2.5‰. Considering that analytical uncertainties are generally <±0.2‰ (2σ), these intergrain and intragrain variations likely represent real oxygen isotopic heterogeneity. In contrast, zircon from the biotite granite shows generally lower $\delta^{18}O_{Zrc}$ values and limited variation between cores (+6.7–7.4‰; mean = +7.0‰) and rims (+6.3‰ and 7.2‰; mean = +6.8‰) (Figure 6c,d; Table S2). The core–rim variation of each grain is insignificant within a range of less than 0.4‰.

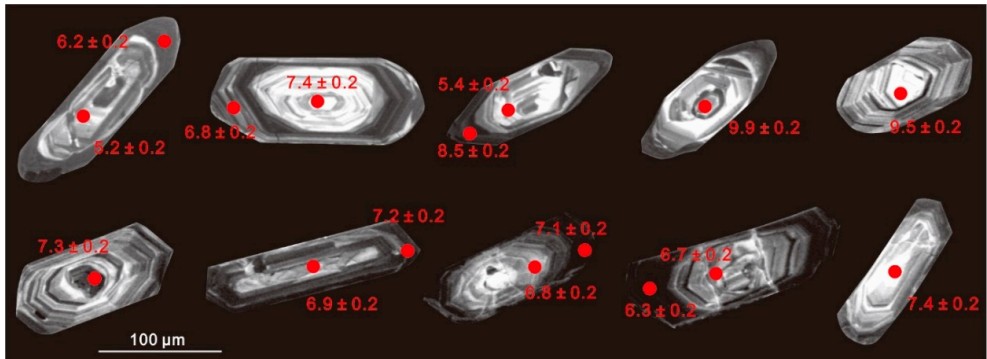

**Figure 5.** Cathodoluminescence (CL) images of representative zircons from the two-mica granite (first row) and the biotite granite (second row). Circles indicate the location of ion microprobe analysis spots, $\delta^{18}O$ values are beside each circle.

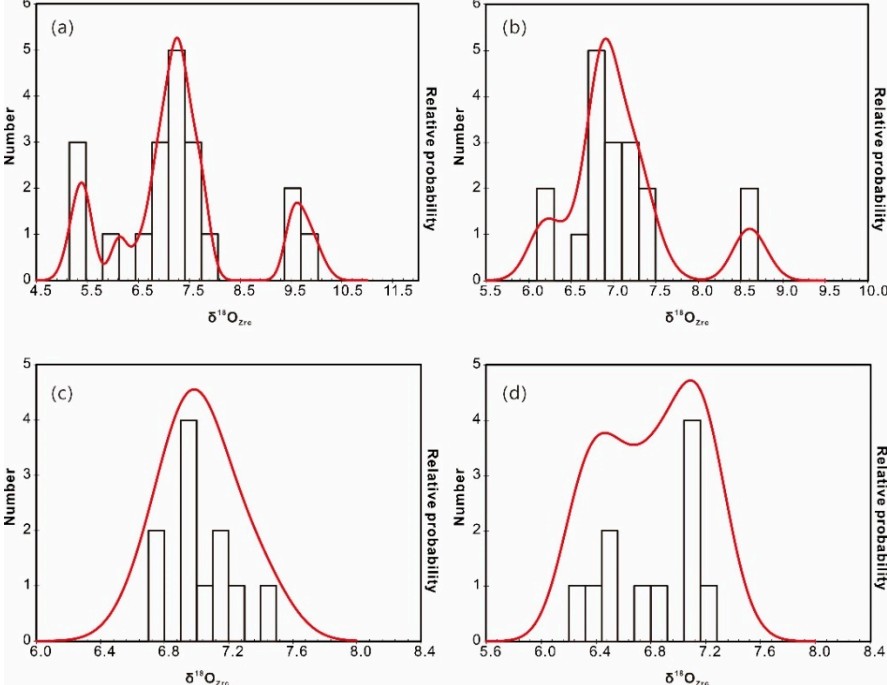

**Figure 6.** Cumulative probability histograms of $\delta^{18}O_{Zrc}$ of the zircon cores (**a**) and rims (**b**) from the two-mica granite, and zircon cores (**c**) and rims (**d**) from the biotite granite.

### 4.3. Hafnium Isotopes in Zircon

Zircons in the two-mica granite show a large range of [176]Hf/[177]Hf, from 0.28219 to 0.28250, corresponding to $\varepsilon_{Hf(t)}$ values ranging from −11.7 to −1.0, with two-stage Hf model ages from 1464 to 2135 Ma. In contrast, zircons from the biotite granite show less variation in [176]Hf/[177]Hf ratios (0.28238 to 0.28251) and $\varepsilon_{Hf(t)}$ values (from −0.4 to −5.1), with more limited two-stage Hf model ages from 1427 to 1724 Ma (Figure 7; Table S4).

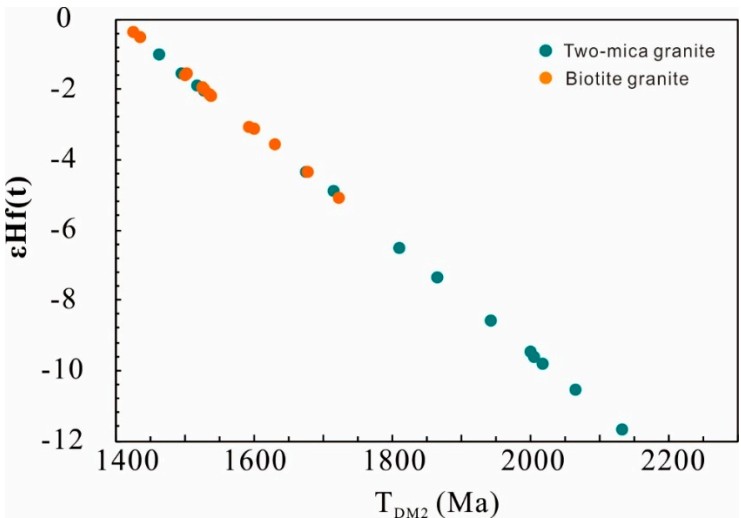

**Figure 7.** $T_{DM2}$ vs. $\varepsilon_{Hf}(t)$ of zircons from the Nashwaak Granite.

## 5. Discussion

### 5.1. Crystal Fractionation and Contamination

Trace elements incorporated into zircon reflect the composition of host magma in which it is growing. Crystal fractionation produces residual melts with relatively higher U and Th content; consequently zircon in such melts evolves toward high Th and U content and low Th/U [27]. Hafnium increases and Zr/Hf decreases in zircon with magma cooling [28–30]. Both the biotite granite and two-mica granite in the Nashwaak Granite have similar U and Th content (Table S3), but zircon rims in the two-mica granite have a wider range of Th/U and Hf content. Relative enrichment in HREE over middle rare earth elements (MREE) in zircons is interpreted as a result of mineral (garnet, hornblende, titanite, and apatite) fractionation in a closed system (Table S3, [17]). In the Nashwaak Granite, only apatite typically occurs as inclusions within biotite. The plot of Th/U versus Yb/Gd may differentiate crystal fractionation and mixing/contamination ([17], Figure 2d). For example, zircon rims of the biotite granite follow the crystal fractionation curve, whereas zircon rims of the two-mica granite are distributed along a mixing curve where Th/U is higher than 0.2 (Table S3). When Th/U is lower than 0.2, zircon rims of two-mica granite plot on the crystal fractionation trend, but some of these grains have relatively high Ti contents (corresponding to high temperatures; [31]), which contradicts a gradual cooling process during fractional crystallization. Thus, zircons with low Th/U and high Ti may also be formed by crustal contamination.

### 5.2. Titanium-In-Zircon Thermometry

The titanium-in-zircon thermometer [28,31,32] provides zircon crystallization temperatures in host melts [17]. Two key parameters of this thermometer are the activities of $SiO_2$ ($aSiO_2$) and $TiO_2$ ($aTiO_2$) in the magma. Model temperatures are calculated for magmas with titanite, in which case a value of 0.7 is adopted for $aTiO_2$, as suggested by [17] and [33]. Variation of 0.1 in $aTiO_2$ leads to changes of 25 to 30 °C in the calculated temperatures. For the Nashwaak Granite, the $aSiO_2$ is 1.0 due

to the presence of quartz, and *a*TiO$_2$ is 0.6, based on the absence of titanite (0.6 is the minimum for felsic magmas). According to this model, the Ti contents of magmatic rims from the biotite granite broadly crystallized at 710 to 850 °C, whereas those of the two-mica granite mainly crystallized between 750 and 800 °C (Figure 8; Table S3).

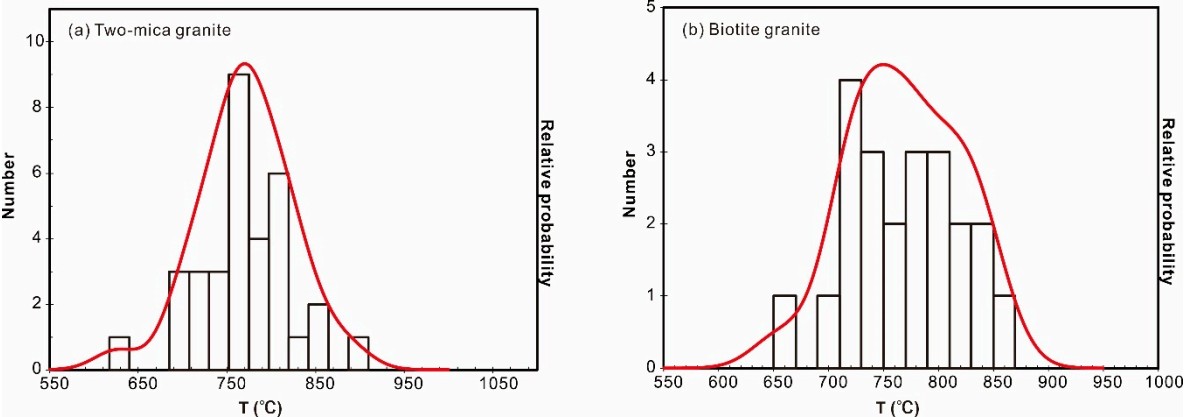

**Figure 8.** Cumulative probability histograms of the Ti-in-zircon thermometer [31] for the two-mica granite (**a**) and biotite granite (**b**).

*5.3. Oxygen Fugacity*

Variation of Ce and Eu anomalies in zircon (Figure 2) reflects the oxidation state of magmas [17]. Cerium and Eu have two valence states. Compared with Ce$^{3+}$, Ce$^{4+}$ is incorporated preferentially into the Zr$^{4+}$ site of zircon; whereas Eu$^{2+}$ is easily accommodated into the Ca$^{2+}$ site of plagioclase. An increase in the oxidation state of magma enhances the positive Ce anomaly, but weakens the negative Eu anomaly in zircon [34].

The Eu/Eu* of zircon rims in the biotite granite generally decreases along with increasing Hf content and decreasing Th/U (Figure 2). Although the influence of oxygen fugacity cannot be totally precluded, the enhancement of a negative Eu anomaly might be dominantly controlled by plagioclase fractional crystallization [30]. Zircon rims in the two-mica granite generally have a Eu/Eu* broadly lower than 0.4, but some scattered values are in the range of 0.6 to 1.8 (Figure 2b). In this case, crustal contamination as discussed above may play a vital role in evolution of the two-mica granite.

An alternative method to evaluate the oxidation state of magma uses the zircon Ce anomaly. In order to avoid error in the estimation of Ce/Ce* arising from low La and Pr content in zircon, the lattice-strain model [24] was used to calculate Ce/Ce*, corresponding Ce$^{4+}$/Ce$^{3+}$ [35], and oxygen fugacity [36] of the magmas. The zircon rims from the biotite granite have Ce$^{4+}$/Ce$^{3+}$ in the range of 1.5 to 44.6 (mean 17.5), with calculated *f*O$_2$ of 10$^{-21}$ to 10$^{-10}$ (mean 10$^{-15}$ bars) (Figure 9), whereas the zircon rims from the two-mica granite are relatively reduced with Ce$^{4+}$/Ce$^{3+}$ of 1.7 to 24.4 (mean 10.1) and calculated *f*O$_2$ of 10$^{-23}$ to 10$^{-13}$ (mean 10$^{-17}$) bars (Figure 9; Table S3). An evolution of *f*O$_2$ with progressive crystallization in two-mica granite is not observed, whereas the *f*O$_2$ of biotite granite increases with decreasing Th/U. This oxidation state could be attributed to H$_2$ degassing [37] or the result of reduction of sulfate to sulfur dioxide during separation of a sulfur-rich magmatic–hydrothermal fluid [38].

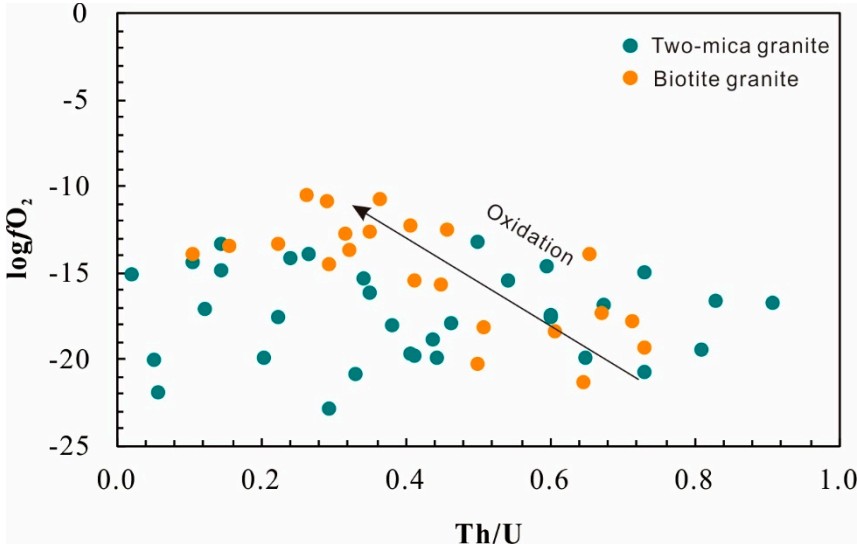

**Figure 9.** Oxygen fugacity variation during magma evolution (Th/U) reflected by the composition of zircon from the Nashwaak Granite.

### 5.4. Magma Sources and Evolution as Recorded by O–Hf Isotopes

Oxygen isotope fractionation between zircon and quartz is a function of temperature and is independent of oxygen fugacity or pressure (Equation (1)):

$$\delta^{18}O_{Qtz} - \delta^{18}O_{Zrc} = \Delta_{Qtz\text{-}Zrc} \approx 1000 \ln(\alpha_{Qtz\text{-}Zrc}) = A_{Qtz\text{-}Zrc}\, 10^6/T^2 \tag{1}$$

where $A_{Qtz\text{-}Zrc} = 2.33 \pm 2.04$ [39] or 2.64 [40] and $T$ = temperature in Kelvin.

If quartz and zircon are in isotopic equilibrium at a particular temperature, then their $\delta^{18}O$ values should lie along a straight line in a plot of $\delta^{18}O_{Zrc}$ vs. $\delta^{18}O_{Qtz}$ (see Figure 5b in [11]). In the biotite granite, the $\delta^{18}O_{Zrc}$ value of the magmatic rim is broadly homogenous (from +6.3‰ to +7.2‰). Consequently, the zircon and quartz in the biotite granite might have formed in a closed system and equilibrated with their hosting magma during crystallization. The average $\delta^{18}O$ values of zircon rims and quartz are +6.8‰ and +9.7‰, respectively. The value of $\Delta_{Qtz\text{-}Zrc}$ is 2.9‰ and the calculated temperature is 688 °C (Tables S1 and S2; [40]) or 675 °C [39].

In the two-mica granite, the zircon has heterogeneous cores with the $\delta^{18}O_{Zrc}$ ranging from +5.2‰ to +9.9‰, indicating various magmatic sources. The zircon rims have $\delta^{18}O_{Zrc}$ in the range of 6.2‰ to 8.7‰ with a mean value of 7.1‰. The average O-isotope fractionation between zircon rims and quartz yields an equilibration temperature below 640 °C [40] or 627 °C [39], much lower than the Ti-in-zircon temperatures. Furthermore, the 2.5‰ difference of $\delta^{18}O_{Zrc}$ in the zircon rims indicates that this magma crystallized in an open system.

Oxygen isotope disequilibrium between zircon and quartz could be caused by several processes, as suggested by [16] for granitoids elsewhere. Feldspar and quartz are less refractory minerals in granites compared to zircon, therefore their oxygen isotope compositions could be readily reset by hydrothermal alteration and recrystallization [15,41,42]. However, large-scale or intense hydrothermal alteration is not evident in the Nashwaak Granite, as indicated by the quartz texture in the CL images. Although quartz grains in biotite granite rarely have dilatant fractures along which hydrothermal alteration could occur (Figure 4c), transects across whole magmatic quartz grains do not show any $\delta^{18}O_{Qz}$ variation, suggesting that hydrothermal alteration was not a main process causing oxygen isotope disequilibrium between quartz and zircon.

Magma mixing (e.g., [43–45]), and crustal assimilation (e.g., [14–16,46,47]) are other processes that can change the $\delta^{18}O$ composition of melts. As shown in [48], early-crystallizing zircon has lower $\delta^{18}O$ than late-crystallized garnet, which formed after contamination of the granitic magma by assimilation

of sedimentary rocks. Supracrustal contamination in the two-mica granite is indicated by the presence of high $\delta^{18}O$ (>+8‰) zircon cores. Such a contamination process could affect the composition and oxygen fugacity of melts. That the $\delta^{18}O$ of magmatic quartz in the two-mica granite did not record the input of supracrustal materials might be due to very late stage crystallization of quartz, after the magma was completely homogenized (see [43]).

Various magma sources can be identified by the oxygen isotope compositions of zircon. Magmatic zircons equilibrated with mantle-derived magmas have average $\delta^{18}O_{Zrc}$ value of +5.3 ± 0.6‰ (2σ; [49]). Notable deviations of $\delta^{18}O_{Zrc}$ from the mantle value are the result of intracrustal recycling. High-$\delta^{18}O$ magmas (+8‰ to above +10‰) reflect assimilation of supracrustal rocks that previously interacted with low temperature fluids [50], while low $\delta^{18}O$ magmas (lower than ca. +4‰) reflect assimilation of supracrustal rocks that previously interacted with meteoric water or seawater at high temperatures [46]. For the Nashwaak Granite, zircon in the biotite granite is in high temperature equilibrium with the coexisting quartz and crystallized in a closed system, thus hydrothermal alteration or assimilation of crustal rock can be ruled out during magma evolution. The homogeneous $\delta^{18}O_{Zr}$ within single grain or between different zircon rims, relatively homogeneous $\varepsilon_{Hf}(t)$ (from −0.39 to −5.10) and "I–type" whole-rock geochemistry characteristics [8] indicate that the biotite granite might be dominantly derived by partially melting of meta-igneous rocks in the lower crust of Ganderia.

The two-mica granite has heterogeneous zircon cores (+5.2‰ to +9.9‰), indicating multiple magma sources. Most $\delta^{18}O_{Zrc}$ of the zircon cores in the two-mica granite are the same as those of the biotite granite. Three cores with $\delta^{18}O_{Zrc}$ values of +5.23‰ to +5.45‰ represent a crustal mafic rock source instead of a juvenile mantle source because their $^{206}Pb/^{238}U$ ages are older than 1.0 Ga. Zircon cores with $\delta^{18}O_{Zrc}$ of +9.51‰ to +9.94‰ are the result of assimilation of supracrustal rocks. This is supported by the feldspar Pb isotopic compositions of two-mica granite, which plot along or near the upper crust reference curve ($^{206}Pb/^{204}Pb$ vs. $^{207}Pb/^{204}Pb$) and on or near the orogene reference curve ($^{206}Pb/^{204}Pb$ vs. $^{208}Pb/^{204}Pb$) (see [51]). The biotite granite has $\varepsilon_{Hf}(t)$ of −0.39 to −5.10 and older $T_{DM2}$ ages of 1427 to 1724 Ma, while the two-mica granite has more negative $\varepsilon_{Hf}(t)$ (as low as −11.7) and older $T_{DM2}$ ages (<2135 Ma). This isotopic difference, caused by crustal assimilation, indicates that an "inverted" crustal structure (low $\varepsilon_{Nd}(t)$ sedimentary rocks overlying more positive $\varepsilon_{Nd}(t)$ crust), similar to that invoked by [4] for the Ganderia in Newfoundland.

## 6. Conclusions

This study indicates the in situ analysis of trace elements of zircon as well as oxygen isotope of both zircon and quartz is an effective way to distinguish crustal contamination from heterogeneity of magmatic source. Some Pridoli to Early Lochkovian granites in Ganderia, proposed to be formed by partial melting of heterogeneous source material (e.g., [52]), might not always be the case, such as the Nashwaak Granite investigated by this work. In situ Hf and O isotope measurements in zircon, the first such data reported for granites in New Brunswick's Central Plutonic Belt, record the detailed magmatic sources and petrogenetic processes of the Nashwaak Granite in the Miramichi Highlands. This granite is dominantly derived from partial melting of Ganderian lower crust and contamination from various supracrustal rocks. The unexposed basement has $\delta^{18}O_{Zrc}$ values between +6.7‰ and +7.4‰, as reflected by inherited zircon cores in the biotite granite. The supracrustal rocks, including mafic rocks with $\delta^{18}O_{Zrc}$ of +5.2‰ to +5.5‰ and sedimentary materials with $\delta^{18}O_{Zrc}$ of +9.5‰ to +9.9‰, were incorporated into a magma derived from the lower basement. Assimilation of those materials led to a decrease in the oxygen fugacity of the hybrid magmas, as indicated by trace element compositions of zircon.

**Supplementary Materials:** The following are available online at http://www.mdpi.com/2075-163X/10/7/614/s1, Table S1: Oxygen isotope of magmatic quartz from the Nashwaak Granitic suite, Table S2: Oxygen isotope of zircon from the Nashwaak Granitic suite, Table S3: Trace element composition of zircon in the Nashwaak Granite, Table S4: Hafnium isotope composition of zircon in the Nashwaak Granite.

**Author Contributions:** Conceptualization, D.R.L. and W.Z.; methodology, D.R.L.; software, W.Z.; validation, D.R.L., W.Z., and K.G.T.; formal analysis, W.Z.; investigation, D.R.L.; resources, K.G.T.; data curation, W.Z.; writing—original draft preparation, W.Z.; writing—review and editing, W.Z.; visualization, W.Z.; supervision, D.R.L.; project administration, D.R.L.; funding acquisition, D.R.L. All authors have read and agreed to the published version of the manuscript.

**Funding:** This research was funded by the New Brunswick Department of Natural Resources (134404-46-01) and NSERC Discovery grant (217095).

**Acknowledgments:** Support by Charlie Morrissy and Justin Bernard during field work was greatly appreciated. Thanks to Geodex Minerals Ltd. for allowing access to their property and information. Edits from Reginald A. Wilson and Christopher J. Eastoe significantly improved this manuscript.

**Conflicts of Interest:** The authors declare no conflict of interest.

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
