# Peer review of "Petrogenesis of the Nashwaak Granite, West-Central New Brunswick, Canada: Evidence from Trace Elements, O and Hf Isotopes of Zircon, and O Isotopes of Quartz"

_minerals, doi:10.3390/min10070614_

Round 1

Reviewer 1 Report

Congratulations on a well-written, concise, and interesting paper. I learned a  lot about oxygen isotopic compositions and trace element compositions in petrogenesis!  As well, I learned much about the basic geology of New Brunswick.

I found certain parts of your paper a little "choppy" in English presentation, but not so much as to require revision. In line 92 you should add the word "was"  i.e.. the sentence needs a verb.

Author Response

added 'was' in line 92

Reviewer 2 Report

The manuscript presents new data on zircon trace element and Hf-isotope and quartz O-isotopes for the Nashwaak Granite. The results are nicely present and the figures are of high quality. The discussion and conclusions are supported by the data. Overall the manuscript can be published with minimal revision.

My only comment is that the authors can elaborate more on the discussion. After obtaining all the new data, the only conclusion reached is that the biotite granite and two-mica granite were "derived from the same basement, but with supracrustal contamination“, which is really not a surprise for granites. Perhaps the authors can discuss more on the physicochemical conditions of the granitic magma genesis?

In the Introduction section, the authors have described three hypotheses of the regional tectonic events associated with the granite genesis (e.g., crustal delamination after closure of the Tetagouche-Exploits backarc basin). However, this has not been followed up in the discussion section, which is a bit disappointing. It would be nice if the authors can also discuss on that.

Figure 5. the μ in the figure is strangely flipped over, flip it back.

Otherwise the paper is fit to go.  

Author Response

  1. Physicochemical conditions of the granitic magma genesis have been discussed enough  we think.
  2. We discussed the possible tectonic setting before but was rejected by Chemical Geology and Lithos, the reviewers disagree my data can solve tectonic controversies, thus, we deleted that section and submitted to Minerals.
  3. Fig. 5 was corrected.

Reviewer 3 Report

Comments on minerals-842078-peer-review-v1 entitled ‘Petrogenesis of the Nashwaak Granite, west-central New Brunswick, Canada: evidence from trace elements, O and Hf isotopes of zircon and O isotope of quartz’ coauthored by Zhang et al., submitted to Minerals

Zhang et al. present a new dataset of in-situ trace elements, O and Hf isotopes of zircon as well as quartz oxygen isotope on the Nashwaak Granite from the New Brunswick Appalachians, suggesting that they may have been I- and/or transitional S-I-types derived dominantly from crustal melts that may have fractionated and been contaminated by supracrustal materials to some extent. This provides insight into the controversy regarding the granite petrogenesis. Thus, this paper is worthy of being published by the journal given few minors are fixed or explained via a minor revision.  

  1. Lines 181 to 191: It is better to explain how to calculate ‘two-stage Hf model age’, and invoke the references to support it. It is noted that the two-mica granite and biotite granite have distinctly different Hf model ages; the former has 1464 to 2135 Ma, where the latter 1054 to 1262 Ma. Such differences strongly suggest that these granites originated from different sources in the crust. This evidence appears not support the conclusion that the two-mica granite and biotite have been derived from the same basement (see the Abstract).
  2. Figure 7: The data points seems not consistent with the text descriptions (also Lines 181 to 191). Check the data and replot this figure.
  3. An explanation is required for why the calculated temperatures based on the quartz-zircon oxygen isotopic fractionation are significantly lower than those calculated from the Ti-in-zircon thermometry. This is a very intriguing problem, indeed! It is likely related to the difference in O and Ti diffusivity in zircon, worthy of further study.
  4. Figure 1b: Is it possible to label where the two-mica granite and biotite granite in the Nashwaak Granite pluton are in the figure (map)? Such information would help understand the field relationships of the composite pluton and the petrogenetic modeling, i.e. process control vs. source control on the geochemical and isotope data of zircon. I suggest that the authors refer some of the important references in dealing with these problems (Yang et al., 2008, Lithos, v. 104, p. 355–377; Clemens, 2018, Contributions to Mineralogy and Petrology, v. 173, p. 93).

Detailed comments are marked on the edited PDF manuscript for authors’ consideration to improve the presentation of this study.

Finally, I would like to see this paper published in the journal because it is a great contribution to the understanding of granite petrogenesis.

Author Response

  1. The age difference of inherited zircon cores is caused by crustal contamination.
  2. changed the data in the text.
  3. We are not able to solve this problem. It's interesting though.
  4. The boundary between the two-mica granite and biotite granite is unclear. just know the northside is two-mica granite and the southside is biotite granite.
